# Bias-Reduced Uncertainty Estimation for Deep Neural Classifiers

**Yonatan Geifman**
Computer Science Department
Technion – Israel Institute of Technology
`yonatan.g@cs.technion.ac.il`

**Guy Uziel**
Computer Science Department
Technion – Israel Institute of Technology
`uzielguy@gmail.com`

**Ran El-Yaniv**
Computer Science Department
Technion – Israel Institute of Technology
`rani@cs.technion.ac.il`

## Abstract

We consider the problem of uncertainty estimation in the context of (non-Bayesian) deep neural classification. In this context, all known methods are based on extracting uncertainty signals from a trained network optimized to solve the classification problem at hand. We demonstrate that such techniques tend to introduce biased estimates for instances whose predictions are supposed to be highly confident. We argue that this deficiency is an artifact of the dynamics of training with SGD-like optimizers, and it has some properties similar to overfitting. Based on this observation, we develop an uncertainty estimation algorithm that selectively estimates the uncertainty of highly confident points, using earlier snapshots of the trained model, before their estimates are jittered (and way before they are ready for actual classification). We present extensive experiments indicating that the proposed algorithm provides uncertainty estimates that are consistently better than all known methods.

## 1 Introduction

The deployment of deep learning models in applications with demanding decision-making components such as autonomous driving or medical diagnosis hinges on our ability to monitor and control their statistical uncertainties. Conceivably, the Bayesian framework offers a principled approach to infer uncertainties from a model; however, there are computational hurdles in implementing it for deep neural networks (Gal & Ghahramani, 2016). Presently, practically feasible (say, for image classification) uncertainty estimation methods for deep learning are based on signals emerging from standard (non Bayesian) networks that were trained in a standard manner. The most common signals used for uncertainty estimation are the raw *softmax response* (Cordella et al., 1995), some functions of softmax values (e.g., entropy), signals emerging from embedding layers (Mandelbaum & Weinshall, 2017), and the MC-dropout method (Gal & Ghahramani, 2016) that proxies a Bayesian inference using dropout sampling applied at test time. These methods can be quite effective, but no conclusive evidence on their relative performance has been reported. A recent NIPS paper provides documentation that an ensemble of softmax response values of several networks performs better than the other approaches (Lakshminarayanan et al., 2017).

In this paper, we present a method of confidence estimation that can consistently improve all the above methods, including the ensemble approach of Lakshminarayanan et al. (2017). Given a trained classifier and a confidence score function (e.g., generated by softmax response activations), our algorithm will learn an improved confidence score function for the same classifier. Our approach is based on the observation that confidence score functions extracted from ordinary deep classifiers tend to wrongly estimate confidence, especially for highly confident instances. Such erroneous estimates constitute a kind of artifact of the training process with an *stochastic gradient descent* (SGD) based optimizers. During this process, the confidence in "easy" instances (for which we expect prediction

with high confidence) is quickly and reliably assessed during the early SGD epochs. Later on, when the optimization is focused on the "hard" points (whose loss is still large), the confidence estimates of the easy points become impaired.

Uncertainty estimates are ultimately provided in terms of probabilities. Nevertheless, as previously suggested (Geifman & El-Yaniv, 2017; Mandelbaum & Weinshall, 2017; Lakshminarayanan et al., 2017), in a non-Bayesian setting (as we consider here) it is productive to decouple uncertainty estimation into two separate tasks: *ordinal ranking* according to uncertainty, and *probability calibration*. Noting that calibration (of ordinal confidence ranking) already has many effective solutions (Naeini et al., 2015; Platt et al., 1999; Zadrozny & Elkan, 2002; Guo et al., 2017), our main focus here is on the core task of ranking uncertainties. We thus adopt the setting of Lakshminarayanan et al. (2017), and others (Geifman & El-Yaniv, 2017; Mandelbaum & Weinshall, 2017), and consider uncertainty estimation for classification as the following problem. Given labeled data, the goal is to learn a pair $(f, \kappa)$, where $f(x)$ is a classifier and $\kappa(x)$ is a confidence score function. Intuitively, $\kappa$ should assign lower confidence values to points that are misclassified by $f$, relative to correct classifications (see Section 2 for details).

We propose two methods that can boost known confidence scoring functions for deep neural networks (DNNs). Our first method devises a selection mechanism that assigns for each instance an appropriate early stopped model, which improves that instance's uncertainty estimation. The mechanism selects the early-stopped model for each individual instance from among snapshots of the network's weights that were saved during the training process. This method requires an auxiliary training set to train the selection mechanism, and is quite computationally intensive to train. The second method approximates the first without any additional examples. Since there is no consensus on the appropriate performance measure for scoring functions, we formulate such a measure based on concepts from selective prediction (Geifman & El-Yaniv, 2017; Wiener & El-Yaniv, 2015). We report on extensive experiments with four baseline methods (including all those mentioned above) and four image datasets. The proposed approach consistently improves all baselines, often by a wide margin. For completeness, we also validate our results using probably-calibrated uncertainty estimates of our method that are calibrated with the well-known Platt scaling technique (Platt et al., 1999) and measured with the negative log-likelihood and Brier score Brier (1950).

## 2 PROBLEM SETTING

In this work we consider uncertainty estimation for a standard supervised multi-class classification problem. We note that in our context *uncertainty* can be viewed as negative *confidence* and vice versa. We use these terms interchangeably. Let $\mathcal{X}$ be some feature space (e.g., raw image pixels) and $\mathcal{Y} = \{1, 2, 3, \ldots, k\}$, a label set for the classes. Let $P(X, Y)$ be an unknown source distribution over $\mathcal{X} \times \mathcal{Y}$. A classifier $f$ is a function $f : \mathcal{X} \to \mathcal{Y}$ whose *true risk* w.r.t. $P$ is $R(f|P) = E_{P(X,Y)}[\ell(f(x), y)]$, where $\ell : Y \times Y \to \mathbb{R}^+$ is a given loss function, for example, the 0/1 error. Given a labeled set $S_m = \{(x_i, y_i)\}_{i=1}^m \subseteq (\mathcal{X} \times \mathcal{Y})$ sampled i.i.d. from $P(X, Y)$, the *empirical risk* of the classifier $f$ is $\hat{r}(f|S_m) \triangleq \frac{1}{m} \sum_{i=1}^m \ell(f(x_i), y_i)$. We consider deep neural classification models that utilize a standard softmax (last) layer for multi-class classification. Thus, for each input $x \in \mathcal{X}$, the vector $f(x) = (f(x)_1, \ldots, f(x)_k) \in \mathbb{R}^k$ is the softmax activations of the last layer. The model's predicted class $\hat{y} = \hat{y}_f(x) = \operatorname{argmax}_{i \in \mathcal{Y}} f(x)_i$.

Consider the training process of a deep model $f$ through $T$ epochs using any mini-batch SGD optimization variant. For each $1 \leq i \leq T$, we denote by $f^{[i]}$ a snapshot of the partially trained model immediately after epoch $i$. For a multi-class model $f$, we would like to define a *confidence score* function, $\kappa(x, i, |f)$, where $x \in \mathcal{X}$, and $i \in \mathcal{Y}$. The function $\kappa$ should quantify confidence in predicting that $x$ is from class $i$, based on signals extracted from $f$. A $\kappa$-score function should induce a partial order over points in $\mathcal{X}$, and thus is not required to distinguish between points with the same score. For example, for any softmax classifier $f$, the vanilla confidence score function is $\kappa(x, i|f) \triangleq f(x)_i$ (i.e., the softmax response values themselves). Perhaps due to the natural probabilistic interpretation of the softmax function (all values are non-negative and sum to 1), this vanilla $\kappa$ has long been used as a confidence estimator. Note, however, that we are not concerned with the standard probabilistic interpretation (which needs to be calibrated to properly quantify probabilities (Guo et al., 2017)).

An *optimal* $\kappa$ (for $f$) should reflect true loss monotonicity in the sense that for every two labeled instances $(x_1, y_1) \sim P(X, Y)$, and $(x_2, y_2) \sim P(X, Y)$,

$$\kappa(x_1, \hat{y}_f(x)|f) \leq \kappa(x_2, \hat{y}_f(x)|f) \iff \mathbf{Pr}_P[\hat{y}_f(x_1) \neq y_1] \geq \mathbf{Pr}_P[\hat{y}_f(x_2) \neq y_2]. \tag{1}$$

## 3 Performance Evaluation of Confidence Scores by Selective Classification

In the domain of (deep) uncertainty estimation there is currently no consensus on how to measure performance (of ordinal estimators). For example, Lakshminarayanan et al. (2017) used the Brier score and the negative-log-likelihood to asses their results, while treating $\kappa$ values as absolute scores. In Mandelbaum & Weinshall (2017) the area under the ROC curve was used for measuring performance. In this section we propose a meaningful and unitless performance measure for $\kappa$ functions, which borrows elements from other known approaches.

In order to define a performance measure for $\kappa$ functions, we require a few concepts from *selective classification* (El-Yaniv & Wiener, 2010; Wiener & El-Yaniv, 2011). As noted in (Geifman & El-Yaniv, 2017), any $\kappa$ function can be utilized to construct a selective classifier (i.e., a classifier with a reject option). Thus, selective classification is a natural application of confidence score functions based on which it is convenient and meaningful to assess performance.

The structure of this section is as follows. We first introduce the (well known) terms *selective classifier*, *selective risk* and *coverage*. Then we introduce the *risk-coverage curve*. We propose to measure the performance of a $\kappa$ function as the area under the risk-coverage curve (AURC) of a selective classifier induced by $\kappa$. The proposed measure is a normalization of AURC where we subtract the AURC of the best $\kappa$ in hindsight. The benefit of the proposed normalization is that it allows for meaningful comparisons accross problems. We term the this normalized metric "excess AURC" (E-ARUC) and it will be used throughout the paper for performance evaluation of $\kappa$ functions.

A *selective classifier* is a pair $(f, g)$, where $f$ is a classifier, and $g : \mathcal{X} \to \{0, 1\}$ is a *selection function*, which serves as a binary qualifier for $f$ as follows,

$$(f, g)(x) \triangleq \begin{cases} f(x), & \text{if } g(x) = 1; \\ \text{reject}, & \text{if } g(x) = 0. \end{cases}$$

The performance of a selective classifier is quantified using *coverage* and *risk*. *Coverage*, defined to be $\phi(f, g) \triangleq E_P[g(x)]$, is the probability mass of the non-rejected region in $\mathcal{X}$. The selective risk of $(f, g)$ is

$$R(f, g) \triangleq \frac{E_P[\ell(f(x), y)g(x)]}{\phi(f, g)}.$$

These two measures can be empirically evaluated over any finite labeled set $S_m$ (not necessarily the training set) in a straightforward manner. Thus, the *empirical selective risk* is,

$$\hat{r}(f, g|S_m) \triangleq \frac{\frac{1}{m} \sum_{i=1}^{m} \ell(f(x_i), y_i)g(x_i)}{\hat{\phi}(f, g|S_m)}, \tag{2}$$

where $\hat{\phi}$ is the *empirical coverage*, $\hat{\phi}(f, g|S_m) \triangleq \frac{1}{m} \sum_{i=1}^{m} g(x_i)$. The overall performance *profile* of a family of selective classifiers (optimized for various coverage rates) can be measured using the *risk-coverage curve* (RC-curve), defined to be the selective risk as a function of coverage.

Given a classifier $f$ and confidence score function $\kappa$ defined for $f$, we define an empirical performance measure for $\kappa$ using an independent set $V_n$ of $n$ labeled points. The performance measure is defined in terms of the following selective classifier $(f, g)$ (where $f$ is our given classifier), and the selection functions $g$ is defined as a threshold over $\kappa$ values, $g_\theta(x|\kappa, f) = \mathbb{1}[\kappa(x, \hat{y}_f(x)|f) > \theta]$.

Let $\Theta$ be the set of all $\kappa$ values of points in $V_n$, $\Theta \triangleq \{\kappa(x, \hat{y}_f(x)|f) : (x, y) \in V_n\}$; for now we assume that $\Theta$ contains $n$ unique points, and later we note how to deal with duplicate values.

The performance of $\kappa$ is defined to be the *area under the (empirical) RC-curve* (AURC) of the pair $(f, g)$ computed over $V_n$,

$$\text{AURC}(\kappa, f|V_n) = \frac{1}{n} \sum_{\theta \in \Theta} \hat{r}(f, g_\theta|V_n).$$

Intuitively, a better $\kappa$ will induce a better selective classifier that will tend to reject first the points that are misclassified by $f$. Accordingly, the associated RC-curve will decrease faster (with decreasing coverage) and the AURC will be smaller.

For example, in Figure 1 we show (in blue) the RC-curve of classifier $f$ obtained by training a DNN trained over the CIFAR-100 dataset. The $\kappa$ induced by $f$ is the softmax response confidence score, $\kappa(x) = \max_i f(x)_i$. The RC-curve in the figure is calculated w.r.t. to an independent labeled set $V_n$ of $n = 10,000$ points from CIFAR-100. Each point on the curve is the empirical selective risk (2) of a selective classifier $(f, g_\theta)$ such that $\theta \in \Theta$. As can be seen, the selective risk is monotonically increasing with coverage. For instance, at full coverage $= 1$, the risk is approximately $0.29$. This risk corresponds to a standard classifier (that always predicts and does not reject anything). The risk corresponding to coverage $= 0.5$ is approximately $0.06$ and corresponds to a selective classifier that rejects half of the points (those whose confidence is least). Not surprisingly, its selective risk is significantly lower than the risk obtained at full coverage.

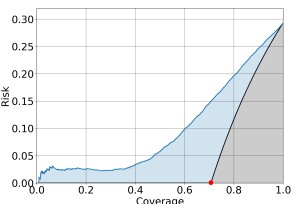

Figure 1: RC-curve for the CIFAR100 dataset with softmax response confidence score. Blue: the RC curve based on softmax response; black: the optimal curve that can be achieved in hindsight.

An optimal in hindsight confidence score function for $f$, denoted by $\kappa^*$, will yield the optimal risk coverage curve. This optimal function rates all misclassified points (by $f$) *lower* than all correctly classified points. The selective risk associated with $\kappa^*$ is thus zero at all coverage rates below $1 - \hat{r}(f|V_n)$. The reason is that the optimal function rejects all misclassified points at such rates. For example, in Figure 1 we show the RC-curve (black) obtained by relying on $\kappa^*$, which reaches zero at coverage of $1 - 0.29 = 0.71$ (red dot); note that the selective risk at full coverage is $0.29$.

Since the AURC of all RC-curves for $f$ induced by any confidence scoring function will be larger than the AURC of $\kappa^*$, we normalize by AURC$(\kappa^*)$ to obtain a unitless performance measure. To compute the AURC of $\kappa^*$, we compute the discrete integral of $\hat{r}$ (w.r.t. $\kappa^*$) from the coverage level of $1 - \hat{r}(f|V_n)$ (0 errors) to 1 ($n\hat{r}$ errors). Thus,

$$\text{AURC}(\kappa^*, f|V_n) = \frac{1}{n} \sum_{i=1}^{\hat{r}n} \frac{i}{n(1-\hat{r})+i}. \tag{3}$$

We approximate (3) using the following integral:

$$\text{AURC}(\kappa^*, f|V_n) \approx \int_0^{\hat{r}} \frac{x}{1-\hat{r}+x} dx = x - (1-\hat{r})\ln(1-\hat{r}+x)\Big|_0^{\hat{r}} = \hat{r} + (1-\hat{r})\ln(1-\hat{r}).$$

For example, the gray area in Figure 1 is the AURC of $\kappa^*$, which equals $0.04802$ (and approximated by $0.04800$ using the integral).

To conclude this section, we define the *Excess-AURC* (E-AURC) as E-AURC$(\kappa, f|V_n) = $ AURC$(\kappa, f|V_n) - $ AURC$(\kappa^*, f|V_n)$. E-AURC is a unitless measure in $[0, 1]$, and the optimal $\kappa$ will have E-AURC $= 0$. E-AURC is used as our main performance measure.

## 4    RELATED WORK

The area of uncertainty estimation is huge, and way beyond our scope. Here we focus only on non-Bayesian methods in the context of deep neural classification. Motivated by a Bayesian approach, Gal & Ghahramani (2016) proposed the Monte-Carlo dropout (MC-dropout) technique for estimating uncertainty in DNNs. MC-dropout estimates uncertainty at test time using the variance statistics extracted from several dropout-enabled forward passes.

The most common, and well-known approach for obtaining confidence scores for DNNs is by measuring the classification margin. When softmax is in use at the last layer, its values correspond to the distance from the decision boundary, where large values tend to reflect high confidence levels. This concept is widely used in the context of classification with a reject option in linear models and in particular, in SVMs (Bartlett & Wegkamp, 2008; Chow, 1970; Fumera & Roli, 2002). In the context of neural networks, Cordella et al. (1995); De Stefano et al. (2000) were the first to propose this approach and, for DNNs, it has been recently shown to outperform the MC-dropout on ImageNet (Geifman & El-Yaniv, 2017).

A $K$-nearest-neighbors (KNN) algorithm applied in the embedding space of a DNN was recently proposed by Mandelbaum & Weinshall (2017). The KNN-distances are used as a proxy for class-conditional probabilities. To the best of our knowledge, this is the first non-Bayesian method that estimates neural network uncertainties using activations from non-final layers.

A new ensemble-based uncertainty score for DNNs was proposed by Lakshminarayanan et al. (2017). It is well known that ensemble methods can improve predictive performance (Breiman, 1996). Their ensemble consists of several trained DNN models, and confidence estimates were obtained by averaging softmax responses of ensemble members. While this method exhibits a significant improvement over all known methods (and is presently state-of-the-art), it requires substantially large computing resources for training.

When considering works that leverage information from the network's training process, the literature is quite sparse. Huang et al. (2017) proposed to construct an ensemble, composed of several snapshots during training to improve predictive performance with the cost of training only one model. However, due to the use of cyclic learning rate schedules, the snapshots that are averaged are fully converged models and produce a result that is both conceptually and quantitatively different from our use of snapshots before convergence. Izmailov et al. (2018) similarly proposed to average the weights across SGD iterations, but here again the averaging was done on fully converged models that have been only fine-tuned after full training processes. Thus both these ensemble methods are superficially similar to our averaging technique but are different than our method that utilizes "premature" ensemble members (in terms of their classification performance).

## 5 MOTIVATION

In this section we present an example that motivates our algorithms. Consider a deep classification model $f$ that has been trained over the set $S_m$ through $T$ epochs. Denote by $f^{[i]}$ the model trained at the $i$th epoch; thus, $f = f^{[T]}$. Take an independent validation set $V_n$ of $n$ labeled points. We monitor the quality of the softmax response generated from $f$ (and its intermediate variants $f^{[i]}$), through the training process, as measured on points in $V_n$. The use of $V_n$ allows us to make meaningful statements about the quality of softmax response values (or any other confidence estimation method) for unseen test points.

We construct the example by considering two groups of instances in $V_n$ defined by confidence assessment assigned using the softmax response values $f$ gives to points in $V_n$. The **green** group contains the highest (99%-100%) percentile of most confident points in $V_n$, and the **red** group contains the lowest (0%-1%) percentile of least confident points. Although the softmax response is rightfully criticized in its ability to proxy confidence (Gal & Ghahramani, 2016), it is reasonable to assume it is quite accurate in ranking green vs. red points (i.e., a prediction by $f$ regarding a red point is likely to be less accurate than its prediction about a green point).

We observe that the prediction of the green points' labels is learned earlier during training of $f$, compared to a prediction of any red point. This fact is evident in Figure 2(a) where we see the training of $f$ over CIFAR-100. Specifically, we see that the softmax response values of green points stabilize at their maximal values around Epoch 80. We also note that the green points in this top percentile are already correctly classified very early, near Epoch 25 (not shown in the figure). In contrast, red points continue to improve their confidence scores throughout. This observation indicates that green points can be predicted very well by an intermediate model such as $f_{130}$. Can we say that $f_{130}$ can estimate the confidence of green points correctly? Recall from Section 3 that a useful method for assessing the quality of a confidence function is the E-AURC measure (applied over an independent validation set). We now measure the quality of the softmax response of all intermediate classifiers $f^{[i]}$,

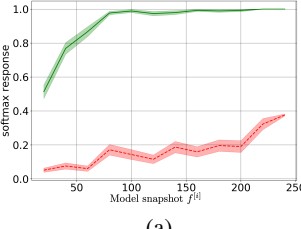 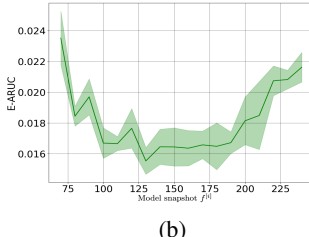 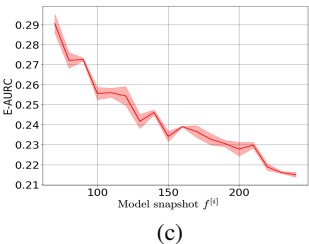

(a)                                  (b)                                  (c)

Figure 2: (a): Average confidence score based on softmax values along the training process. Green (solid): 100 points with the highest confidence; red (dashed): 100 points with the lowest confidence. (b, c): The E-AURC of softmax response on CIFAR-100 along training for 5000 points with highest confidence (b), and 5000 points with lowest confidence (c).

$1 \leq i \leq T$, over the green points and, separately, over the red points. Figure 2(b) shows the E-AURC of the green points. The $x$-axis denotes indices $i$ of the intermediate classifiers ($f^{[i]}$); the $y$-axis is E-AURC($f^{[i]}, \kappa | \{$ green points$\}$). Similarly, Figure 2(c) shows the E-AURC of the red points. We see that for the green points, the confidence estimation quality improves (almost) monotonically and then degrades (almost) monotonically. The best confidence estimation is obtained by intermediate classifiers such as $f_{130}$. Surprisingly, the final model $f^{[T]}$ is one of the worst estimators for green points! In sharp contrast, the confidence estimates for the red points monotonically improves as training continues. The best estimator for red points is the final model $f^{[T]}$. This behavior can be observed in all the datasets we considered (not reported).

The above dynamics indicates that the learning of uncertainty estimators for easy instances conceptually resembles overfitting in the sense that the assessment of higher confidence points in the test set degrades as training continues after a certain point. To overcome this deficiency we propose an algorithm that uses the concept of early stopping in a pointwise fashion, where for each sample (or set of samples) we find the best intermediate snapshot for uncertainty estimation.

## 6   SUPERIOR CONFIDENCE SCORE BY EARLY STOPPING

In this section, first we present a supervised algorithm that learns an improved scoring function for a given pair $(f, \kappa)$, where $f$ is a trained deep neural classifier, and $\kappa$ is a confidence scoring function for $f$'s predictions. In principle, $\kappa : \mathcal{X} \to \mathbb{R}$, where $\kappa(x)$ can be defined as any mapping from the activations of $f$ applied on $x$ to $\mathbb{R}$. All the confidence estimation methods we described above comply with this definition.[1] Our algorithm requires a labeled training sample. The second algorithm we present is an approximated version of the first algorithm, which does not rely on additional training examples.

### 6.1   THE POINTWISE EARLY STOPPING ALGORITHM FOR CONFIDENCE SCORES

Let $f$ be a neural classifier that has been trained using any (mini-batch) SGD variant for $T$ epochs, and let $F \triangleq \{f^{[i]} : 1 \leq i \leq T\}$ be the set of intermediate models obtained during training ($f^{[i]}$ is the model generated at epoch $i$). We assume that $f$, the snapshots set $F$, and a confidence score function for $f$, $\kappa(\cdot, \cdot | f) : \mathcal{X} \to (0, 1]$), are given.[2] Let $V_n$ be an independent training set.

The Pointwise Early Stopping (PES) algorithm for confidence scores (see pseudo-code in Algorithm 1) operates as follows. The pseudo-code contains both the training and inference procedures. At each iteration of the training main loop (lines 3-11), we extract from $V$ (which is initialized as a clone of the set $V_n$) a set of the $q$ most uncertain points. We abbreviate this set by $S$ (the "layer"). The size of the layer is determined by the hyperparameter $q$. We then find the best model in $F$ using the

---

[1]One can view an ensemble as a summation of several networks. MC-dropout can be described as an ensemble of several networks.

[2]We assume that $\kappa$ is bounded and normalized.

---

**Algorithm 1** *The Pointwise Early Stopping Algorithm for Confidence Scores (PES)*

---

1: **function** TRAIN($V_n$,$q$,$\kappa$,$F$)
2:      $V \leftarrow V_n$; $\Theta \leftarrow []$; $K \leftarrow []$; $T \leftarrow |F|$; $j \leftarrow |F|$
3:      **for** $i = 0$ **to** $\lceil n/q \rceil$ **do**
4:          $r \leftarrow \{\kappa(x, y_{f^{[T]}}(x)|f^{[j]}) : (x,y) \in V\}$
5:          $\tilde{\theta} \leftarrow r_{(q)}$                         $\triangleright$ $r_{(q)}$ indicates the $q$th order statistic of $r$
6:          $S \leftarrow \{(x,y)|\kappa(x, \hat{y}_{f^{[T]}}(x)|f^{[j]}) < \tilde{\theta}\}$
7:          $j = \mathrm{argmin}_{0 < j \leq T}(\text{E-AURC}(\kappa(x, \hat{y}_{f^{[T]}}(x)|f^{[j]}), f^{[T]}|S)$
8:          $K[i] \leftarrow \kappa(\cdot, \cdot|f^{[j]})$
9:          $\Theta[i] \leftarrow \max(\{\kappa(x, \hat{y}_{f^{[T]}}(x)|f^{[j]}) : (x,y) \in S\})$
10:         $V \leftarrow V \setminus S$
11:      **end for**
12:      **return** K,$\Theta$
13: **end function**
14: **function** ESTIMATE CONFIDENCE($x$,$f$,$K$,$\Theta$)
15:      **for** $i = 0$ **to** $|K| - 1$ **do**
16:          **if** $\kappa_i(x, \hat{y}_f(x)) \leq \theta_i$ **then**                $\triangleright$ $\kappa_i$ is the $i$th element of $K$
17:              **return** $i + \kappa_i(x, \hat{y}_f(x))$
18:          **end if**
19:      **end for**
20:      **return** $i + \kappa_i(x, \hat{y}_f(x))$
21: **end function**

---

E-AURC measure with respect to $S$. This model, denoted $f^{[j]}$, is found by solving

$$j = \operatorname*{argmin}_{0 < j \leq T}(\text{E-AURC}(\kappa(x, \hat{y}_{f^{[T]}}(x)|f^{[j]}), f^{[T]}|S). \tag{4}$$

The best performing confidence score over $S$, and the threshold over the confidence level, $\theta$, are saved for test time (lines 8-9) and used to associate points with their layers. We iterate and remove layer after layer until $V$ is empty.

Our algorithm produces a partition of $\mathcal{X}$ comprising layers from least to highest confidence. For each layer we find the best performing $\kappa$ function based on models from $F$.

To infer the confidence rate for given point $x$ at test time, we search for the minimal $i$ that satisfies

$$\kappa_i(x, \hat{y}_{f^{[T]}}(x)) \leq \theta_i,$$

where $\kappa_i$ and $\theta_i$ are the $i$'th elements of $K$ and $\Theta$ respectively.

Thus, we return $\kappa(x, \hat{y}_{f^{[T]}}(x)|F) = i + \kappa_i(x, \hat{y}_{f^{[T]}}(x))$, where $i$ is added to enforce full order on the confidence score between layers, recall that $\kappa \in (0, 1]$ .

## 6.2    AVERAGED EARLY STOPPING ALGORITHM

As we saw in Section 6.1, the computational complexity of the PES algorithm is quite intensive. Moreover, the algorithm requires an additional set of labeled examples, which may not always be available. The Averaged Early Stopping (AES) is a simple approximation of the PES motivated by the observation that "easy" points are learned earlier during training as shown in Figure 2(a). By summing the area under the learning curve (a curve that is similar to 2(a)) we leverage this property and avoid some inaccurate confidence assessments generated by the last model alone. We approximate the area under the curve by averaging $k$ evenly spaced points on that curve.

Let $F$ be a set of $k$ intermediate models saved during the training of $f$,

$$F \triangleq \{f^{[i]} : i \in \text{linspace}(t, T, k)\},$$

where linspace$(t, T, k)$ is a set of $k$ evenly spaced integers between $t$ and $T$ (including $t$ and $T$). We define the output $\kappa$ as the average of all $\kappa$s associated with models in $F$,

$$\kappa(x, \hat{y}_f(x)|F) \triangleq \frac{1}{k} \sum_{f^{[i]} \in F} \kappa(x, \hat{y}_f(x), f^{[i]}).$$

As we show in Section 7, AES works surprisingly well. In fact, due to the computational burden of running the PES algorithm, we use AES in most of our experiments below.

## 7 EXPERIMENTAL RESULTS

We now present results of our AES algorithm applied over the four known confidence scores: softmax response, NN-distance (Mandelbaum & Weinshall, 2017), MC-dropout (Gal & Ghahramani, 2016) ans Ensemble (Lakshminarayanan et al., 2017) (see Section 4). For implementation details for these methods, see Appendix A. We evaluate the performance of these methods and our AES algorithm that uses them as its core $\kappa$. In all cases we ran the AES algorithm with $k \in \{10, 30, 50\}$, and $t = \lfloor 0.4T \rfloor$. We experiment with four standard image datasets: CIFAR-10, CIFAR-100, SVHN, and Imagenet (see Appendix A for details).

Our results are reported in Table 4. The table contains four blocks, one for each dataset. Within each block we have four rows, one for each baseline method. To explain the structure of this table, consider for example the 4th row, which shows the results corresponding to the softmax response for CIFAR-10. In the 2nd column we see the E-AURC ($\times 10^3$) of the softmax response itself (4.78). In the 3rd column, the result of AES applied over the softmax response with $k = 10$ (reaching E-AURC of 4.81). In the 4th column we specify percent of the improvement of AES over the baseline, in this case -0.7% (i.e., in this case AES degraded performance). For the imagenet dataset, we only present results for the softmax response and ensemble. Applying the other methods on this large dataset was computationally prohibitive.

| | Baseline | AES ($k = 10$) | | AES ($k = 30$) | | AES ($k = 50$) | |
|---|---|---|---|---|---|---|---|
| | E-AURC | E-AURC | % | E-AURC | % | E-AURC | % |
| **CIFAR-10** | | | | | | | |
| Softmax | 4.78 | 4.81 | -0.7 | **4.49** | **6.1** | 4.49 | 6.0 |
| NN-distance | 35.10 | 5.20 | 85.1 | 4.70 | 86.6 | **4.58** | **86.9** |
| MC-dropout | 5.03 | 5.32 | -5.8 | **4.99** | **0.9** | 5.01 | 0.4 |
| Ensemble | 3.74 | 3.66 | 2.1 | **3.50** | **6.5** | 3.51 | 6.2 |
| **CIFAR-100** | | | | | | | |
| Softmax | 50.97 | 41.64 | 18.3 | 39.90 | 21.7 | **39.68** | **22.1** |
| NN-distance | 45.56 | 36.03 | 20.9 | 35.53 | 22.0 | **35.36** | **22.4** |
| MC-dropout | 47.68 | 49.45 | -3.7 | 46.56 | 2.3 | **46.50** | **2.5** |
| Ensemble | 34.73 | 31.10 | 10.5 | **30.72** | **11.5** | 30.75 | 11.5 |
| **SVHN** | | | | | | | |
| Softmax | 4.24 | **3.73** | **12.0** | 3.77 | 11.1 | 3.73 | 11.9 |
| NN-distance | 10.08 | **7.69** | **23.7** | 7.81 | 22.5 | 7.75 | 23.1 |
| MC-dropout | 4.53 | 3.79 | 16.3 | 3.81 | 15.8 | **3.79** | **16.3** |
| Ensemble | 3.69 | **3.51** | **4.8** | 3.55 | 3.8 | 3.55 | 4.0 |
| **ImageNet** | | | | | | | |
| Softmax | 99.68 | 96.88 | 2.8 | 96.09 | 3.6 | **94.77** | **4.9** |
| Ensemble | 90.95 | **88.70** | **2.47** | 88.84 | 2.32 | 88.86 | 2.29 |

Table 1: E-AURC and % improvement for AES method on CIFAR-10, CIFAR-100, SVHN and Imagenet for various $k$ values compared to the baseline method. All E-AURC values are multiplied by $10^3$ for clarity.

Let us now analyze these results. Before considering the relative performance of the baseline methods compares to ours, it is interesting to see that the E-AURC measure nicely quantifies the difficulty level of the learning problems. Indeed, CIFAR-10 and SVHN are known as relatively easy problems and the E-AURC ranges we see in the table for most of the methods is quite small and similar. CIFAR-100 is considered harder, which is reflected by significantly larger E-AURC values recorded for the various methods. Finally, Imagenet has the largest E-AURC values and is considered to be the

hardest problem. This observation supports the usefulness of E-AURC. A non-unitless measure such as AUC, the standard measure, would not be useful in such comparisons.

It is striking that among all 42 experiments, our method improved the baseline method in 39 cases. Moreover, when applying AES with $k = 30$, it always reduced the E-AURC of the baseline method. For each dataset, the ensemble estimation approach of Lakshminarayanan et al. (2017) is the best among the baselines, and is currently state-of-the-art. It follows that for all of these datasets, the application of AES improves the state-of-the-art. While the ensemble method (and its improvement by AES) achieve the best results on these datasets, these methods are computationally intensive. It is, therefore, interesting to identify top performing baselines, which are based on a single classifier. In CIFAR-10, the best (single-classifier) method is softmax response, whose E-AURC is improved 6% by AES (resulting in the best single-classifier performance). Interestingly, in this dataset, NN-distance incurs a markedly bad E-AURC (35.1), which is reduced (to 4.58) by AES, making it on par with the best methods for this dataset. Turning to CIFAR-100, we see that the (single-classifier) top method is NN-distance, with an E-AURC of 45.56, which is improved by 22% using AES.

| | NLL | | | Brier score | | |
|---|---|---|---|---|---|---|
| | Baseline | AES | % | Baseline | AES | % |
| **CIFAR-10** | | | | | | |
| Softmax | 0.193 | **0.163** | **15.8** | 0.051 | **0.045** | **12.9** |
| NN-distance | 0.211 | **0.166** | **21.1** | 0.055 | **0.045** | **17.0** |
| MC-dropout | 0.208 | **0.196** | **5.74** | 0.059 | **0.058** | **2.03** |
| Ensemble | 0.158 | **0.141** | **10.9** | 0.044 | **0.039** | **10.6** |
| **CIFAR-100** | | | | | | |
| Softmax | 0.539 | **0.430** | **20.2** | 0.178 | **0.137** | **23.0** |
| NN-distance | 0.485 | **0.397** | **18.2** | 0.156 | **0.127** | **18.7** |
| MC-dropout | 0.454 | **0.438** | **3.34** | 0.152 | **0.150** | **1.10** |
| Ensemble | 0.416 | **0.377** | **9.3** | 0.131 | **0.118** | **9.8** |
| **SVHN** | | | | | | |
| Softmax | 0.109 | **0.088** | **19.2** | 0.027 | **0.022** | **17.5** |
| NN-distance | 0.136 | **0.121** | **10.6** | 0.032 | **0.030** | **5.4** |
| MC-dropout | 0.165 | **0.141** | **14.73** | 0.045 | **0.039** | **13.98** |
| Ensemble | 0.092 | **0.082** | **10.39** | 0.023 | **0.021** | **12.14** |
| **Imagenet** | | | | | | |
| Softmax | 0.511 | **0.504** | **1.38** | 0.168 | **0.165** | **1.63** |
| Ensemble | 0.497 | **0.491** | **1.20** | 0.162 | **0.160** | **1.55** |

Table 2: NLL and Brier score of AES method applied with Platt scaling on CIFAR-10, CIFAR-100, SVHN and Imagenet compared to the baseline method (calibrated as well).

Next we examine AES applied together with *probability calibration*. We calibrate the results of the AES algorithm using the Platt scaling technique; see (Platt et al., 1999) for details. Platt scaling is applied on the results of the AES algorithm with $k = 30$, and compared to the *independently scaled* underlying measure without AES. Performance is evaluated using both negative log-likelihood (NLL) and the Brier score (Brier, 1950). For further implementation details of this experiment see Appendix A. The results appear in Table 2. As can easily be seen, the probability scaling results are remarkably consistent with our raw uncertainty estimates (measured with the E-AURC) over all datasets and underlying uncertainty methods. We conclude that AES also improves calibrated probabilities of the underlying uncertainty measures, and the E-AURC can serve as a reliable proxy also for calibrated probabilities.

We implemented the PES algorithm only over the softmax response method (SR) for several datasets. To generate an independent training set, which is required by PES, we randomly split the original validation set (in each dataset) into two parts and took a random 70% of the set for training our algorithm, using the remaining 30% for validation. The reason we only applied this algorithm over softmax responses is the excessive computational resources it requires. For example, when

| Dataset | E-AURC - SR | E-AURC - PES | % Improvement |
|---------|-------------|--------------|---------------|
| CIFAR-10 | $4.6342 \pm 0.07$ | $\mathbf{4.3543 \pm 0.06}$ | **6.04** |
| CIFAR-100 | $51.3172 \pm 0.43$ | $\mathbf{41.9579 \pm 0.39}$ | **18.24** |
| SVHN | $4.1534 \pm 0.18$ | $\mathbf{3.7622 \pm 0.16}$ | **9.41** |
| Imagenet | $97.1393 \pm 0.77$ | $\mathbf{94.8668 \pm 0.85}$ | **2.34** |

Table 3: E-AURC and % improvement for the Pointwise Early Stopping algorithm (PES) compared to the softmax response (SR) on CIFAR-10, CIFAR-100 and SVHN. All E-AURC values are multiplied by $10^3$ for clarity.

applying PES over NN-distance, the time complexity is $nmTk + \mathcal{O}(TC_f(S_m) + TC_f(V_n))$, where $k$ is the number of neighbours and $C_f(S_m)$ is the time complexity of running a forward pass of $m$ samples using the classifier $f$. Similarly, the complexity of PES when the underlying scores are from MC-dropout is $\mathcal{O}(dTC_f(V_n))$ where $d$ is the number of dropout iterations (forward passes) of the MC-dropout algorithm. Thus, when $n = 7000$, $T = 250$ (the parameters used for applying PES over CIFAR-100), and with $d = 100$ (as recommended in Gal & Ghahramani (2016)), this amounts to 175,000,000 forward passes. We set $q = \lfloor n/3 \rfloor$. We repeated the experiment over 10 random training–validation splits and report the average results and the standard errors in Table 3.

As seen, PES reduced the E-AURC of softmax on all datasets by a significant rate. The best improvement was achieved on CIFAR-100 (E-AURC reduced by 18%).

Our difficulties when applying the PES algorithm on many of the underlying confidence methods, and the outstanding results of the AES motivate further research that should lead to improving the algorithm and making it more efficient.

## 8  CONCLUDING REMARKS

We presented novel uncertainty estimation algorithms, which are motivated by an observation regarding the training process of DNNs using SGD. In this process, reliable estimates generated in early epochs are later on deformed. This phenomenon somewhat resembles the well-known overfitting effect in DNNs.

The PES algorithm we presented requires an additional labeled set and expensive computational resources for training. The approximated version (AES) is simple and scalable. The resulting confidence scores our methods generate systematically improve all existing estimation techniques on all the evaluated datasets.

Both PES and AES overcome confidence score deformations by utilizing available snapshot models that are generated anyway during training. It would be interesting to develop a loss function that will explicitly prevent confidence deformations by design while maintaining high classification performance. In addition, the uncertainty estimation of each instance currently requires several forward passes through the network. Instead it would be interesting to consider incorporating *distillation* (Hinton et al., 2015) so as to reduce inference time. Another direction to mitigate the computational effort at inference time is to approximate PES using a single model per instance based on an early stopping criterion similar to the one proposed by Mahsereci et al. (2017).

## ACKNOWLEDGMENTS

This research was supported by The Israel Science Foundation (grant No. 81/017).

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

## A  IMPLEMENTATION AND EXPERIMENTAL DETAILS

### A.1  DATASETS

**CIFAR-10:** The CIFAR-10 dataset (Krizhevsky & Hinton, 2009) is an image classification dataset containing 50,000 training images and 10,000 test images that are classified into 10 categories. The image size is $32 \times 32 \times 3$ pixels (RGB images).

**CIFAR-100:** The CIFAR-100 dataset (Krizhevsky & Hinton, 2009) is an image classification dataset containing 50,000 training images and 10,000 test images that are classified into 100 categories. The image size is $32 \times 32 \times 3$ pixels (RGB images).

**Street View House Numbers (SVHN):** The SVHN dataset (Netzer et al., 2011) is an image classification dataset containing 73,257 training images and 26032 test images classified into 10 classes representing digits. The images are digits of house numbers cropped and aligned, taken from the Google Street View service. Image size is $32 \times 32 \times 3$ pixels.

**ImageNet:** The ImageNet dataset (Deng et al., 2009) is an image classification dataset containing 1300000 color training images and another 50,000 test images classified into 1000 categories.

### A.2  ARCHITECTURES AND HYPER PARAMETERS

**VGG-16:** For the first three datasets (CIFAR10, CIFAR100, and SVHN), we used an architecture inspired by the VGG-16 architecture (Simonyan & Zisserman, 2014). We adapted the VGG-16 architecture to the small images size and a relatively small dataset size based on (Liu & Deng, 2015). We trained the model for 250 epochs using SGD with a momentum value of 0.9. We used an initial learning rate of 0.1, a learning rate multiplicative drop of 0.5 every 20 epochs, and a batch size of 128. A standard data augmentation was used including horizontal flips, rotations, and shifts. In this learning regime, we reached a validation error of 6.4% for CIFAR-10, 29.2% for CIFAR-100 and 3.54% for SVHN.

**Resnet-18:** For ImageNet dataset, we used the Resnet-18 architecture (He et al., 2016); we trained the model using SGD with a batch size of 256 and momentum of 0.9 for 90 epochs. We used a learning rate of 0.1, with a learning rate multiplicative decay of 0.1 every 30 epochs. The model reached a (single center crop) top 1 validation accuracy of 69.6% and top 5 validation accuracy of 89.1%.

## A.3 METHODS IMPLEMENTATIONS AND HYPER PARAMETERS

**Softmax Response:** For the softmax response method (SR) we simply take the relevant softmax value of the sample, $\kappa(x, i|f) = f(x)_i$.

**NN-distance:** We implemented the NN-distance method using $k = 500$ for the nearest neighbors parameter. We didn't implemented the two proposed extensions (embedding regularization, and adversarial training), this add-on will degrade the performance of $f$ for better uncertainty estimation, which we are not interested in. Moreover, running the NN-distance with this add-on will require to add it to all other methods to manage a proper comparison.

**MC-dropout:** The MC-dropout implemented with $p = 0.5$ for the dropout rate, and 100 feed-forward iterations for each sample.

**Ensemble:** The Ensemble method is implemented as an average of softmax values across ensemble of 5 DNNs.

**Platt scaling (Platt et al., 1999):** The Platt scaling is applied as follows. Given a confidence measure $\kappa$ and a validation set $V$, the scaling is the solution of the logistic regression from $\kappa(x, \hat{y}_f(x)|f)$ to $\kappa^*(x, \hat{y}_f(x)|f)$, where $\kappa^*(x, \hat{y}_f(x)|f)$ is defined as 0 when $x \neq \hat{y}_f(x)$ and 1 otherwise. We train the logistic regression models based on all points in $V$. To validate the training of this calibration we randomly split (50-50) the original test set to a training and test subsets. The calibration is learned over the training subset and evaluated on the test subset. The performance of the resulting scaled probabilities has been evaluated using both negative log likelihood (NLL) and the Brier score (Brier, 1950), which is simply the average $L_2$ distance between the predicted and the true probabilities.

## B DETAILED RESULTS

We provide here the table of the experiments of AES for softmax response and NN-distance now with standard errors. Due to computational complexity the standard error for all other methods has not been computed.

| | Baseline | AES ($k = 10$) | | AES ($k = 30$) | | AES ($k = 50$) | |
|---|---|---|---|---|---|---|---|
| | E-AURC | E-AURC | % | E-AURC | % | E-AURC | % |
| **CIFAR-10** | | | | | | | |
| Softmax | $4.78 \pm 0.11$ | $4.81 \pm 0.11$ | -0.7 | $\mathbf{4.49 \pm 0.09}$ | 6.1 | $\mathbf{4.49 \pm 0.08}$ | 6.0 |
| NN-distance | $35.10 \pm 6.54$ | $\mathbf{5.20 \pm 0.28}$ | 85.1 | $\mathbf{4.70 \pm 0.18}$ | 86.6 | $\mathbf{4.58 \pm 0.10}$ | 86.9 |
| **CIFAR-100** | | | | | | | |
| Softmax | $50.97 \pm 0.56$ | $\mathbf{41.64 \pm 1.81}$ | 18.3 | $\mathbf{39.90 \pm 1.61}$ | 21.7 | $\mathbf{39.68 \pm 1.59}$ | 22.1 |
| NN-distance | $45.56 \pm 0.15$ | $\mathbf{36.03 \pm 1.82}$ | 20.9 | $\mathbf{35.53 \pm 1.80}$ | 22.0 | $\mathbf{35.36 \pm 1.85}$ | 22.4 |
| **SVHN** | | | | | | | |
| Softmax | $4.24 \pm 0.11$ | $\mathbf{3.73 \pm 0.05}$ | 12.0 | $\mathbf{3.77 \pm 0.04}$ | 11.1 | $\mathbf{3.73 \pm 0.03}$ | 11.9 |
| NN-distance | $10.08 \pm 1.17$ | $\mathbf{7.69 \pm 0.59}$ | 23.7 | $\mathbf{7.81 \pm 0.47}$ | 22.5 | $\mathbf{7.75 \pm 0.57}$ | 23.1 |

Table 4: E-AURC and % improvement for AES method on CIFAR-10, CIFAR-100, SVHN and ImageNET for various $k$ values compared to the baseline method. All E-AURC values are multiplied by $10^3$ for clarity.

## C MOTIVATION - EXTENDED EXPERIMENTS

In Section 5 we motivated our method by dividing the domain $\mathcal{X}$ to "easy points" (green) and "hard points" (red). We demonstrated that the "easy points" have a phenomenon similar to overfitting, where at some point during training the E-AURC measured for "easy points" start degrading. This observation strongly motivates our strategy that extracts information from early stages of the training process that helps to recover uncertainty estimates of the easy points. Here, we extend this demonstration that previously was presented done with respect to the softmax $\kappa$ function. In Figures 3 and

4 we show plots similar to Figure 2(b,c) for the MC-dropout and NN-distance, respectively. It is evident that the overfitting occurs in all cases. but to a much lesser extent in the case of MC-dropout. This result is consistent with the results of the AES algorithm where E-AURC improvement over the MC-dropout was smaller compared to the improvements achieved for the other two methods. In the case of NN-distance a slight overfitting also affects the easy points, but the hard instances are affected much more severely. Thus, from this perspective in all three cases the proposed correction stratgey is potentially useful.

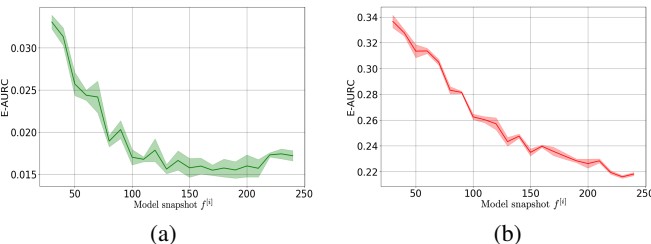

Figure 3: The E-AURC of MC-dropout on CIFAR-100 along training for 5000 points with highest confidence (a), and 5000 points with lowest confidence (b).

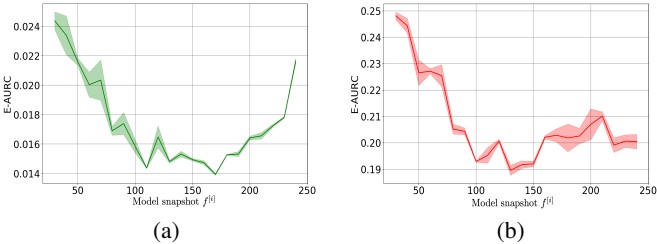

Figure 4: The E-AURC of NN-distance on CIFAR-100 along training for 5000 points with highest confidence (a), and 5000 points with lowest confidence (b).

