# OpenReview forum: "Bias-Reduced Uncertainty Estimation for Deep Neural Classifiers"
_ICLR.cc/2019/Conference_

### Official Review · AnonReviewer1 · 2018-10-31
**Non-bayesian uncertainty estimation for deep nets.**

**Rating:** 7
**Confidence:** 2

**Review:**

In this papers, the authors introduce a new technique to output uncertainty estimates from any family of neural nets. The key insight in this paper is that when considering existing SGD methods the following behavior occurs: if we think of "easy" and "hard" to classify datapoints, a NN trained with SGD will output good uncertainty estimates early on in training, but once the network focusses on tuning the parameters for the hard cases, the uncertainty estimates for the easy datapoints deteriorates. The algorithms proposed by the authors takes an existing uncertainty method (or confidence score function) and uses intermediate snapshots of SGD training to improve the final uncertainty estimates. Note that the focus in this work is on ranking uncertainties (and the authors suggest to leave calibrating uncertainties to existing methods).

The paper generally is well written (e.g. section 5) although I found section 3 to be a bit hard to follow. I'm not very familiar with the area itself but I was surprised to see in Section 7 that the results are not compared to full Bayesian methods (possibly on a dataset that lends itself well to that).

Notes:
- Section 3, "A selective classifier ..." -> I think this section could use some additional untuition to make the explanation more understandable.
- Section 3, "defined to be the selective risk as a function of coverage." -> do you mean as a sequence of functions g?
-

---

> ### Author Response · Authors · 2018-11-19
> **Working on a revised version**
>
> Thanks for your comments. We are working on several improvements that address all your comments and will upload a new version of the paper by the end of this week.
> In particular, we are streamlining Section 3, as requested, and we clarify your question on the risk-coverage curve -  you are correct, the risk coverage curve measures the tradeoff with respect to several g functions optimized for various coverage rates.

---

> ### Author Response · Authors · 2018-11-25
> **Revised version**
>
> We uploaded a revised version of the paper. The main changes/additions are:
>
> * Calibration - We applied the well-known Platt scaling to calibrate the output of the AES algorithm. Results are added in Table 2. Calibrated AES outperforms all contenders.
>
> * Use of other evaluation metrics - The calibration experiment is evaluated using both log-likelihood and Brier score.
>
> * Extension of Figure 2 to other methods - In Appendix C we present analogous graphs to Figure 2 for the MC-dropout and NN-distance uncertainty estimators. These new graphs are qualitatively similar to the graphs already presented for softmax response, but are somewhat less pronounced.
>
> * Clarification of section 3 - We added a high level overview of the developments in this section and added a walk-through of the risk-coverage curve in Figure 1.
>
> * Notation change - f_i changed to f^{[i]}.
>
> * Figure 2(a) - adapted to be color-blind friendly.
>
> * All spotted typos and minor comments fixed.

---

### Official Review · AnonReviewer2 · 2018-11-01
**Good paper but contribution feels isolated from related work**

**Rating:** 7
**Confidence:** 3

**Review:**

-- Paper Summary --

The proposed methodology draws on the connection between boosting in ensemble learning and SGD for training DNNs, whereby misclassified instances are implicitly targeted in later training iterations once easier examples have been classified correctly. The authors observe that this incurs a trade-off in which easily-classified examples become susceptible to overfitting at later stages in the training procedure when the network parameters adapt to fit more complex examples. Two early stopping algorithms are proposed in order to mitigate this issue. The first approach, PES, is more robust, but too computationally expensive to be applied in practice; on the other hand, AES approximates the former procedure by directly assuming that easier training examples will be learnt earlier on in the training procedure. The proposed technique is shown to calibrate the confidence scores obtained from state-of-the-art approaches for training deep nets, resulting in substantial performance improvements with respect to the proposed E-AURC metric.

-- General Commentary --

- The paper isolates itself from other post-calibration methods by stating that ‘our focus here is only on the core task of ranking uncertainties’. In doing so, there is no comparison to other calibration methods, which makes it difficult to properly assess the impact of this work in comparison to other papers addressing the poor calibration of uncertainty typically associated with deep nets. The authors immediately dismiss PES as being too computationally expensive, so I’d be interested in at least seeing AES be compared to more lightweight calibration methods.

 - This paper champions the use of an alternative metric (E-AURC) for assessing model quality, which is the sole quantity of interest in the experimental evaluation. While the E-AURC metric is indeed well-motivated in Section 3, I could see there being some scepticism as to why more traditional metrics such as log likelihood aren’t used here. This would also facilitate comparison to other post-calibration methods. In this regard, the authors should consider supplementing their experiments with more widely-used metrics not limited to uncertainty ranking.

- I would be interested in seeing the analysis shown in Figure 2 extended to each of the baseline models discussed in the paper. Such examples would give a clearer perspective of which methods are particularly susceptible to the overfitting problem targeted by the methodology proposed in this work.

- Some of the notation in the problem statement is a bit confusing, with i being simultaneously  used as the training iteration number as well as an index for Y. This needs to be updated.

- There’s a lot of whitespace in Figure 1 which could be avoided by giving additional examples of how the metric works.

- ‘Early Stopping without a Validation Set (Mahsereci et al, 2017)’ warrants a citation here.

- The paper is otherwise generally well-written and a pleasure to read. Some spotted typos:

P1: for highly confident instance(s)
P3: which borrows element(s)
P3: ‘unit-less’ : this is unhyphenated in another part of the text
P5: Final reference to Figure 2(b) should refer to Figure 2(c) instead
P7: which (is) initialized

-- Recommendation --

I admit to feeling fairly ambivalent about this paper - on one hand, the paper is well-written and its contributions are effectively communicated. While myopic, the experiments also convincingly showcase the performance improvements obtained by applying AES over the baseline methods. On the downside, this paper limits itself to comparing the proposed approaches to baseline methods where no other calibration is carried out. Lack of direct comparison against other post-calibration methods results in the paper adding little to the overall literature on DNNs other than asserting that calibration through early stopping is better than not doing anything else.

Pros/Cons:

+ Properly-motivated contributions and well-written paper.
+ The two early stopping algorithms are explained well, even if the appealing connection to boosting gets lost somewhere along the way.
+ Results show that AES improves the results of several DNN training approaches.

- Use of E-AURC as the sole metric for assessing quality in the Experiments section exposes this paper to instant criticism.
- The notion of preserving model snapshots can be problematic when training requires thousands of epochs.
- No comparison to other post-calibration techniques.


** Post-rebuttal

Score increased to a 7 following rebuttal and paper revision.

---

> ### Author Response · Authors · 2018-11-19
> **Thanks for your comments.**
>
> Thanks for your comments. We are working on several improvements that address all your comments and will upload a new version of the paper by the end of this week.
> In the meantime, here is a summary of our observations from experiments designed to address your concerns.
>
> Calibration-
> In response to your concerns, we have trained Platt scaling calibration over the AES outputs. We already completed Cifar-10 and Cifar-100 runs.
> With the exception of MC-dropout, also the calibrated AES improves the calibration without it. After completing all runs, we will add everything to the paper (end of the week).
>
> Evaluation metrics-
> We are including both negative log-likelihood and the Brier score to evaluate the post calibration results. Here again, not all experiments are done, but from Cifar-10/100 these results are consistent with the E-AURC pre-calibration evaluation. All will be added to the paper.
>
> All your other smaller comments have already been addressed and will appear in the new version soon.

---

> ### Author Response · Authors · 2018-11-25
> **Revised version**
>
> We uploaded a revised version of the paper. The main changes/additions are:
>
> * Calibration - We applied the well-known Platt scaling to calibrate the output of the AES algorithm. Results are added in Table 2. Calibrated AES outperforms all contenders.
>
> * Use of other evaluation metrics - The calibration experiment is evaluated using both log-likelihood and Brier score.
>
> * Extension of Figure 2 to other methods - In Appendix C we present analogous graphs to Figure 2 for the MC-dropout and NN-distance uncertainty estimators. These new graphs are qualitatively similar to the graphs already presented for softmax response, but are somewhat less pronounced.
>
> * Clarification of section 3 - We added a high level overview of the developments in this section and added a walk-through of the risk-coverage curve in Figure 1.
>
> * Notation change - f_i changed to f^{[i]}.
>
> * Figure 2(a) - adapted to be color-blind friendly.
>
> * All spotted typos and minor comments fixed.

---

> ### Comment · AnonReviewer2 · 2018-11-26
> **Post-rebuttal Update**
>
> I thank the authors for taking the time to properly improve upon the issues highlighted in the reviews - the addition of Platt scaling and inclusion of more evaluation metrics also makes the Experiments section more convincing. I believe the story told by the paper feels more complete now, and I am duly raising my score up to a 7.

---

### Official Review · AnonReviewer3 · 2018-11-08
**Bias-reduced uncertainty estimation for deep neural classifiers**

**Rating:** 7
**Confidence:** 4

**Review:**

This paper presents an improved method for uncertainty estimation in deep neural networks, based on  their observations that the confidence scores based on highly confident points and low confidence points would be quite different.

The paper is in general well presented. The proposed method is well motivated (as in section 5). The results of the AES algorithm support well the proposed idea, which nevertheless looks simple.

Section 3 needs further improvement in clarity.

Figure 1 needs to be better presented.

Figure 2(a) - please make the curves color-blind friendly.

SGD (stochastic gradient descent?) needs to be defined, and you can't assume everybody knows what it is.

---

> ### Author Response · Authors · 2018-11-19
> **Working on a revised version**
>
> Thanks for your comments. We are working on several improvements that address all your comments and will upload a new version of the paper by the end of this week.

---

> ### Author Response · Authors · 2018-11-25
> **Revised version**
>
> We uploaded a revised version of the paper. The main changes/additions are:
>
> * Calibration - We applied the well-known Platt scaling to calibrate the output of the AES algorithm. Results are added in Table 2. Calibrated AES outperforms all contenders.
>
> * Use of other evaluation metrics - The calibration experiment is evaluated using both log-likelihood and Brier score.
>
> * Extension of Figure 2 to other methods - In Appendix C we present analogous graphs to Figure 2 for the MC-dropout and NN-distance uncertainty estimators. These new graphs are qualitatively similar to the graphs already presented for softmax response, but are somewhat less pronounced.
>
> * Clarification of section 3 - We added a high level overview of the developments in this section and added a walk-through of the risk-coverage curve in Figure 1.
>
> * Notation change - f_i changed to f^{[i]}.
>
> * Figure 2(a) - adapted to be color-blind friendly.
>
> * All spotted typos and minor comments fixed.

---

### Author Response · Authors · 2018-11-25
**Revised version**

We uploaded a revised version of the paper. The main changes/additions are:

* Calibration - We applied the well-known Platt scaling to calibrate the output of the AES algorithm. Results are added in Table 2. Calibrated AES outperforms all contenders.

* Use of other evaluation metrics - The calibration experiment is evaluated using both log-likelihood and Brier score.

* Extension of Figure 2 to other methods - In Appendix C we present analogous graphs to Figure 2 for the MC-dropout and NN-distance uncertainty estimators. These new graphs are qualitatively similar to the graphs already presented for softmax response, but are somewhat less pronounced.

* Clarification of section 3 - We added a high level overview of the developments in this section and added a walk-through of the risk-coverage curve in Figure 1.

* Notation change - f_i changed to f^{[i]}.

* Figure 2(a) - adapted to be color-blind friendly.

* All spotted typos and minor comments fixed.

---

### Public Comment · (anonymous) · 2018-12-10
**A question about the cost of the AES**

Thanks for your great work. Your paper is well-written and presents very interesting insights and solutions. Could you clarify the cost of the AES for me? For the AES with k=50, does it means we will need to save 50 models and do 50 forward pass to get the uncertainty?

---

> ### Author Response · Authors · 2018-12-15
> **Answer about the cost of the AES**
>
> Thanks for your comment. Your are correct. Indeed, when inference cost is a concern, we recommend using the PES algorithm which is more expensive to train, but much cheaper at test time. An open direction is distillation [1] of AES to a single, fast model (see concluding remarks in our paper).
>
> [1] - Geoffrey Hinton, Oriol Vinyals, and Jeff Dean. Distilling the knowledge in a neural network. arXiv preprint arXiv:1503.02531, 2015.

---

> > ### Public Comment · (anonymous) · 2019-01-15
> > **Question on the implementation of AES with ensemble**
> >
> > Congrats for the acceptance! According to the description in paper, I assume when applying AES on an ensemble of 5 with k=10, every member in the ensemble has 10 snapshots and 50 forward passes are needed in total? Is this correct? Thanks!

---

### Meta-Review · Area_Chair1 · 2018-12-12
**Interesting idea**

**Confidence:** 4
**Recommendation:** Accept (Poster)

**Metareview:**

The paper proposes an improved method for uncertainty estimation in deep neural networks.

Reviewer 2 and AC note that the paper is a bit isolated in terms of comparing the literature.

However, as all of reviewers and AC found, the paper is well written and the proposed idea is clearly new/interesting.